# Aujeszky’s Disease in South-Italian Wild Boars (*Sus scrofa*): A Serological Survey

**DOI:** 10.3390/ani11113298

**Published:** 2021-11-18

**Authors:** Gianmarco Ferrara, Consiglia Longobardi, Filomena D’Ambrosi, Maria Grazia Amoroso, Nicola D’Alessio, Sara Damiano, Roberto Ciarcia, Valentina Iovane, Giuseppe Iovane, Ugo Pagnini, Serena Montagnaro

**Affiliations:** 1Department of Veterinary Medicine and Animal Productions, University of Naples “Federico II”, Via Delpino no. 1, 80137 Naples, Italy; gianmarco.ferrara@unina.it (G.F.); f.dambrosi91@gmail.com (F.D.); sara.damiano@unina.it (S.D.); rciarcia@unina.it (R.C.); iovane@unina.it (G.I.); upagnini@unina.it (U.P.); 2Department of Mental, Physical Health and Preventive Medicine, University of Campania “Luigi Vanvitelli”, Naples, Largo Madonna delle Grazie 1, 80138 Napoli, Italy; consiglia.longobardi@unicampania.it; 3Department of Animal Health, Istituto Zooprofilattico Sperimentale del Mezzogiorno, Via Salute, n. 2 Portici, 80055 Naples, Italy; mamoroso@izsmportici.it (M.G.A.); nicola.dalessio@cert.izsmportici.it (N.D.); 4Department of Pharmacy, University of Salerno, Fisciano, 84084 Salerno, Italy; viovane@unisa.it

**Keywords:** Aujeszky’s disease, ELISA, serosurvey, Southern Italy, wild boars

## Abstract

**Simple Summary:**

Aujeszky’s disease (AD, pseudorabies) is a viral disease of suids caused by Suid Herpesvirus 1 (SHV-1) also referred as Aujeszky’s disease virus (ADV) or Pseudorabies virus (PRV). The aim of our study was to evaluate seroprevalence of AD in wild boar hunted in the Campania Region, during the 2016–2017 hunting season. A total of 503 serum samples from wild boars hunted in the provinces of Campania Region were tested for antibody against ADV using an ELISA assay. A Seroprevalence of 23.85% (120/503, 95% Confidence Interval (CI 95%: 20.15–27.55) was found. Gender was not significantly associated with of ADV seropositivity (*p* > 0.05), while the presence of ADV antibodies was statistically associated with age (>36-month, *p* < 0.0001) and location (Avellino, *p* = 0.0161).

**Abstract:**

Aujeszky’s disease (AD, pseudorabies) is a viral disease of suids caused by Suid Herpesvirus 1 (SHV-1) also referred as Aujeszky’s disease virus (ADV) or Pseudorabies virus (ADV). Domestic pig and Wild boar (*Sus scrofa*) are the natural host, but many species can be infected with ADV. The aim of our study was to evaluate seroprevalence of AD in wild boar hunted in the Campania Region, during the 2016–2017 hunting season. A total of 503 serum samples from wild boars hunted in the provinces of Campania Region (Southern Italy) were collected and were tested for antibody against ADV using an AD, blocking ELISA assay. A Seroprevalence of 23.85% (120/503, 95% Confidence Interval (CI): 20.15–27.55) was found. Gender was not significantly associated with of ADV seropositivity (*p* > 0.05), while the presence of ADV antibodies was statistically associated with age (>36-month, *p* < 0.0001) and location (Avellino, *p* = 0.0161). Our prevalence values are like those obtained in 2010 in our laboratory (30.7%), demonstrating a constant circulation of ADV in the area.

## 1. Introduction

Pseudorabies (PR), also known as Aujeszky’s disease (AD), is a viral disease that affects Suidae and causes significant damage to the pig industry worldwide, with increased mortality in piglet and abortion in sows, while adult pigs usually heal after infection [1,2]. The severity of the disease depends on age of the host and the virulence of the viral strain involved in the outbreak.

The causative agent of AD is Suid Herpevirus 1 (SHV-1) also known as Aujeszky’s disease (ADV), or Pseudorabies virus (PRV), a neurotropic virus, which belongs to the genus Varicellovirus, subfamily Alphaherpesvirinae, family Herpesviridae. Its host range is wide and includes several species that are members of the suidae family, both domestic and wild animals, which are the unique hosts able to develop a productive infection and can act as reservoirs for ADV [3]. In fact, suidae are the only known reservoirs and natural hosts of Aujeszky’s disease, while several mammals are susceptible to infections as blind-end hosts [3]. Concern on ADV infection in humans as arises, due to occasional controversial cases of encephalitis described in the literature, diagnosed on the basis of clinical symptoms, history of contact with animals and/or detection of antibodies [4,5,6]. The virus, similarly to other Herpesvirus, can establish a latent infection in pigs, the reactivation of which can occur after natural stimuli or because of stress factors [3].

As is known, wild animals can be reservoirs and spreaders of infections to livestock [7], in this context the wild boar-domestic pig interface represents an example of this interaction. In fact, wild and domestic suidae are affected by infectious and parasitic diseases that can shift from one species to the other [1]. AD is one of these infectious diseases, and this infection in wild boars is widespread worldwide with different prevalence rates [8]. However, additional information is needed to better understand the actual risk posed by latent infected wild swine as amplifiers of multi-host pathogens that may affect native wildlife as well as domestic animals.

In Italy and in all EU countries, an eradication plan is in place, including large-scale mandatory vaccination with delete-gE vaccines (first vaccination between 60 and 90 days of age), application of additional guarantees for pig movements between Member States [9] with increased efforts to control the disease. Pigs can be dispatched from any Member State to any other if the conditions laid down in Decision 2008/185/EC [9] are respected. These conditions are under the control of the Member State of origin and are more stringent for movements of pigs to a Member State with a higher animal health status [9]. This strategy has led to a decrease in the incidence and prevalence of the pseudorabies in Italy and in several European Union (EU) Member States [10].

In Italy the national AD control program, which provides for mandatory vaccination with delete-gE vaccines was started in 1997 [11] and together with the community provisions in force about AD eradication [9] resulted in a significant reduction in the incidence and prevalence of the disease, but AD has not yet been eradicated [12].

In fact, currently, in the Campania region, the prevalence of AD at herd and animal level is 0.5% and 0.05%, respectively; unfortunately, the ADV seems to be still present in feral pigs [13].

In Italy, there are no national plans for the control and/or eradication of pseudorabies in wild boars. The Italian regions establish programs and draw up regional plans for the surveillance of infectious diseases of wildlife, based on the characteristics of the territory.

The presence of wild suidae can compromise the success of the disease eradication plans [14]. In fact, wild boars may be maintenance or spill-over hosts in the AD epidemiology and can reintroduce infection to domestic pigs. Consequently, the control of infectious diseases in species, such as the Eurasian wild boars (*Sus scrofa*) is desirable, especially in areas such as Europe where, in recent years, this species has spread [7,8,9,10,11,12,13,14,15]. In fact, in recent decades, the European wild boar population has been raised steadily [16,17,18]; also in Italy, the number of wild boars has significantly increased with a consequent expansion of the colonized areas. This scenario is due to the modification of the habitat of this species, in fact, wild boars are now present in the plains, on the hills and in the mountainous areas and the presence of wild animals was also observed on the outskirts of urban areas [19].

The growth of the wild boar population has led to an increase in the probability of contact between wild and domestic pigs with a consequent increase in the possible transmission of infectious diseases between the two species.

Italy is extremely rich in biodiversity; it has the highest number and density of animal and plant species within the European Union, as well as a high rate of endemism. The extension of the Italian territory over about 10° of latitude and the simultaneous presence of coastal, hilly, and mountainous environments determined the presence of ecological niches, close in space but very different from each other.

In this scenario, Campania is one of the Italian regions with the highest biodiversity. In fact, about 27% of the region is included among the protect areas, for a total of 367,548 hectares. The fauna is characterized by marten (Martes martes), Eurasian badger (Meles meles), red fox (Vulpes vulpes), few wolves (Canis lupus), and especially wild boars.

For this reason, we have selected the Campania for our serosurvey. In fact, the aim of this study was to determine the seroprevalence of AD in wild boars in the Campania region of Italy and compare our results with its correlation with age, gender, and location of hunting and with results obtained in our previous study in the same area in 2010 [13].

Risk factors such as age, gender, and location have been chosen to consider the ethology of wild boar. Age, gender, and location are closely linked with geographical density of animals, which represents the major risk factor associated with ADV [20]. In fact, wild boars are naturally social animals, living in female-dominated sounders consisting of barren sows and mothers with young led by an old matriarch. Male boars leave their sounder at about a year of age, while females either remain with their mothers or establish new territories nearby. Subadult males may live in loosely knit groups, while adult and old males tend to be isolated outside the breeding season [21].

## 2. Materials and Methods

### 2.1. Ethical Approval

Approval from the ethics committee was not required since no live animals were used for this study. In fact, the serum samples used in our work were provided by hunters with appropriate wild boar hunting licenses.

### 2.2. Study Area and Sampling

The Campania region is located on the south-western portion of the Italian Peninsula, with an area of 13,590 km^2^ (41°00′00″ N–14°30′00″ E) and a coastline of 350 km (217 mi) on the Tyrrhenian Sea. The region has a temperate Mediterranean climate along the coast and continental in the inland area.

Pigs are one of the most raised animals in the Campania region, in fact, there are about 22,200 pig farms, with a total of about 175,000 animals. About 96.5% of farms are small family farms, while fattening and breeding farms represent 2% and 1.5%, respectively [22].

The survey was conducted on 503 wild boar serum samples during the hunting season of 2016–2017.

A non-probabilistic sampling method (convenience sampling) was used in this study. Serum samples were taken from animals hunted in the wooded areas of municipalities belonging to the provinces of Avellino, Salerno, Benevento and Caserta, considering the hunting tradition of the individual areas.

The animals were hunted according to standard wild boar hunting procedures and transported to a central processing site for cleaning, slaughtering and sampling procedures. We collected 1–3 cm3 of whole blood via heart puncture or cranial sinus puncture. Blood samples were centrifuged at 1100× *g* for 15 min and frozen at −80 °C until testing [23]. For each sample, gender, age, hunting site, and date of collection were recorded. The age of the hunted wild boars was established by a tooth eruption pattern and the animals were classified into three age ranges: yearling (0–12 months); sub-adults (13–36 months); adults (>36 months) [24].

### 2.3. Serological Assay

The presence of antibodies against ADV was investigated using a commercial competitive enzyme-linked-immunosorbent assay (IDEXX ADV/ADV gB Antibody test kit), performed according to the manufacturer’s instructions. The ELISA ADV/ADV gB Antibody test kit used ADV antigen-coated microwell plates and assessed the competition between antibodies present in the serum samples tested and a monoclonal anti-ADV gB monoclonal antibody conjugated to horseradish peroxidase (HRP). The absorbance at 650 nm, A (650), was measured using Glomax Multi Detection System multiwell plate reader (Promega Corporation, Madison, WI, USA).

### 2.4. Statistical Analysis

MedCalc Statistical Software version 16.4.3 (MedCalc Software, Ostend, Belgium; www.medcalc.org; accessed on 19 January 2021) was used for statistical analysis. To compare the proportions of positivity relative to the dependent variables and to establish statistical significance within each class (age, gender, and location), the chi-square test was used.

Binary logistic models were applied to the variables (age, gender, location) associated with the seroprevalence for ADV using JMP version 14.1.0 (SAS Institute Inc.). *p* < 0.05 was considered significant. Odds ratios (OR) and their 95% confidence intervals (CI) were calculated to quantify the significant differences between the categories.

## 3. Results

In our study, 503 wild boar serum samples were tested, collected in 119 hunting areas in the 4 provinces of the Campania Region (Figure 1).

The age group between 13 and 36 months (44.3%) was the more represented, followed by the age classes 0–12 (31.6%) and >36 (24.1%). There were 302 males (60%) and 201 females (40%). The province of Salerno provided the largest number of samples (61.6%, *n* = 310), followed by Avellino (20.4%, *n* = 103), Benevento (15.5%, *n* = 78) and Caserta (2.4%, *n* = 12).

A total of 120 out of 503 serum samples tested (23.85%; 95% Confidence Intervals (CI) 20.15–27.55) were positive to competitive ELISA for ADV antibodies (Table 1). The univariate and multivariate analyses results showed a statistical association between age (Chi-square test: DF = 2, *n* = 503, *p*-value = 0.000069) and location (Chi-square test: DF = 3, *n* = 503, *p*-value = 0.0161 (Table 1). In particular, wild boars over 36 months of age showed a higher probability of ADV seropositivity than younger animals with an odds ratio (OR) of 3.2 (CI 95% 1.8–5.7) and 1.9 (CI 95% 1.2–3.2), respectively, versus 0–12 and 13–36 months old.

A risk of ADV seropositivity was significantly correlated with the Avellino Province (34.0%, CI 95% 25.2–42.9) with OR of 1.7 (CI 95% 1.04–2.8), 2.5 (CI 95% 0.5–12.3), and 3.1 (CI 95% 1.5–6.7), respectively, versus Salerno, Caserta, and Benevento Provinces (Table 1).

## 4. Discussion

Pseudorabies is an endemic disease affecting wild boars in many parts of the world [25,26]. The increase in the wild animal population, such as wild boars, in many parts of Europe represents a risk factor for diseases shared between wildlife and livestock [27]. Consequently, wildlife health surveillance has become a problem of national and international interest, in particular regarding infectious diseases under eradication and control plans, such as ADV, and involving both wild animals and livestock.

The seroprevalence of pseudorabies in wild boars was 23.85% (120/503, 95% CI: 20.15–27.55), our data should be compared with the value of a similar survey conducted in Campania during the hunting season 2005–2006 (30.7%) showing a slight decrease in the prevalence of pseudorabies in the wild swine population [13]. Previous surveys conducted in Italy show that the ADV prevalence found in our study is slightly lower than the value found in studies performed in Tuscany (28.60%) [28], in the North-West regions of Italy (30.39%) [29] and central Italy (30.9%) [30]. Taken together, these data show that ADV seroprevalence is roughly constant in Italian wild boar population for a long time [13,28,29,30] and that ADV is more common in this species than we thought.

The overall ADV seroprevalence of 23.85% (95% CI: 20.15–27.55) in wild boar in Campania Region, for the time period 2016–2017 is lower than values reported for ADV in feral swine populations, in other European Country, such us Czech Republic (30%), Slovenia (26%), Greece (35.1%), Spain (from 12.7% to 57.7%), and Turkey (69.9%) [31,32,33,34,35]. On the other hand, lower prevalence values were found in France (from 0% to 18.4%), Poland (8.7%), Germany (12.09%), Belgium (18.1%), and Swiss (0.57%) [36,37,38,39,40].

However, caution is needed when comparing seroprevalence values obtained from different diagnostic tests, with various sampling methods and in several geographic areas. In fact, all these variables can influence the results of the serosurvey and overestimate or underestimate the impact of infectious diseases in the wild fauna.

Univariate and multivariate analyses indicated a correlation between age, location, and positivity to ADV.

Our study a higher infection rates have been found in wild boars over 36 months of age (38%) than in animal of 0–12 (15.1%) and 13–36 (22.4%) months old with an OR versus 36 months of 3.2 and 1.9, respectively. Our results agree with other survey conducted in Italy and in other European country, that show that antibody prevalence in wild swine is correlated with age [10,29,30,32,34,41].

Several surveys have observed a higher ADV exposure in females compared to males [16,17,18,19,20,21,22,23,24,25,26,27,28,29,30,31,32,33], leading us to theorize that spread inside maternal groups occur more frequently. In our study, there is not statistically association between gender and ADV positivity, though female show the highest prevalence value (26.4%). Maybe a bigger sample would give more precise statistical results, but this was not possible in our study, nevertheless a possible explanation of this trend is the change in the composition of the wild boar population due to the hunting pressure, the expansion of the species, and the ethology of the wild boar [15]. In fact, as previously said, in wild boar, the contact between adult females and young animals is more probable than among male adults. Wild boars are typically gregarious swine, living in clusters consisting of a dominant matriarch, sows, and mothers with piglets. Joung male leave the group at 8–15 months old and tend to be isolated out of the breeding season, while females remain with their mothers or create new groups [21].

Furthermore, a high seroprevalence was found in animals living in the province of Avellino (34.0%), followed by Salerno (23.2%), Caserta (16.7%), and Benevento (14.1%). Avellino location is a risk factor for ADV seropositivity with an OR of 1.7, 2.5, and 3.1, respectively, versus Salerno, Caserta, and Benevento province. This scenario reflects the distribution of the wild boars in Campania region, in fact Avellino is the province with the largest population of wild boars, followed by Salerno, Caserta and Benevento [42].

## 5. Conclusions

In conclusion, our data confirms that wild boar populations, in Campania Region, is not free from Aujeszky’s disease. In fact, looks endemic in Campania, in agreement with what has been described in the other region of Italy and in Europe.

Thus, Aujeszky’s disease in wild swine appears to be a source of infection for domestic pigs, as well as domestic and wild carnivores. Therefore, serosurvey to detect specific antibody to ADV, especially for endemic disease, are highly recommended and our data indicates the necessity for the continuance of this type of serosurvey aiming to assess and monitoring the prevalence of ADV infection among wild boars.

## Figures and Tables

**Figure 1 animals-11-03298-f001:**
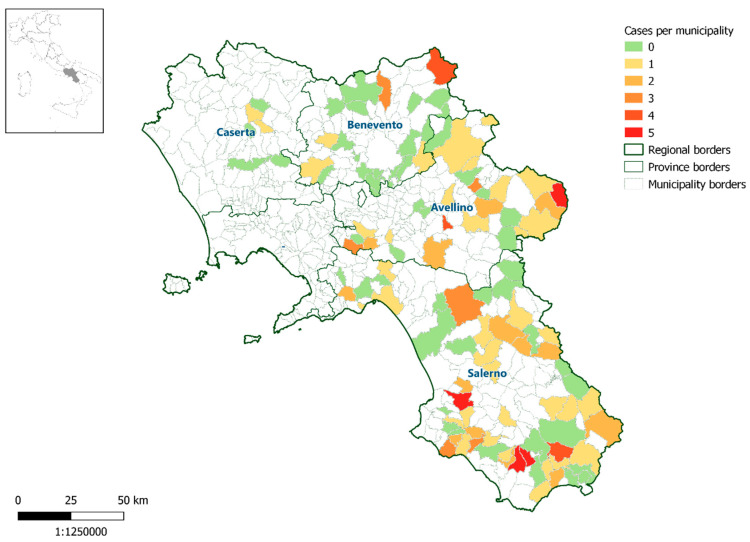
Map of the Campania Region (Italy) showing the distribution of samples of Eurasian wild boar (*Sus scrofa*) seropositive for Pseudorabies virus.

**Table 1 animals-11-03298-t001:** Seroprevalence of infection with ADV and risk factor analysis by age, gender and location in Wild Boars in the Campania Region as detected by competitive ELISA.

Factor	*n*	Elisa Positive	%	ES%	95%CI	χ^2^	*p*	OR	95%CI	*p*
Total	503	120	23.85	3.7	20.15–27.55	-	-	-		
Age										
0–12	159	24	15.1	5.5	9.6–20.6	20.329	0.000069	3.2	1.8–5.7	<0.0001
13–36	223	50	22.4	5.1	17.3–27.5			1.9	1.2–3.2	0.0065
>36	121	46	38.0	8.8	29.2–46.8			Ref		
Gender										
Male	302	67	22.2	4.6	17.6–26.8	0.943	0.3314	0.8	0.5–1.2	0.2814
Female	201	53	26.4	6.04	20.36–32.4					
Province										
Salerno	310	72	23.2	4.7	18.5–27.9	10.306	0.0161	1.7	1.04–2.8	0.0319
Caserta	12	2	16.7	20.6	0−37.3			2.5	0.5–12.3	0.2386
Benevento	78	11	14.1	8.8	5.3–22.9			3.1	1.5–6.7	0.0031
Avellino	103	35	34.0	8.9	25.2–42.9			Ref		

## Data Availability

Not applicable.

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
