# Peer review of "Aujeszky’s Disease in South-Italian Wild Boars (Sus scrofa): A Serological Survey"

_animals, 2021, doi:10.3390/ani11113298_

Round 1

Reviewer 1 Report

  1. It would be better for readers to choose one of the names ADV or PRV in the text. (The overall PRV seroprevalence......./.....n Italy show that the ADV prevalence....). It will be avoid confusion.
  2. Similar results are in this article (A serological survey of selected pathogens in wild boar (Sus scrofa) in northern Turkey). Since Greece and Turkey are neighboring countries, this article can also be cited.

Author Response

Dear Reviewer,

Thank you very much for your suggestions. You will find enclosed the response point by point to your comments, please note that the integrated text will appear underlined inside the manuscript.

Best regards

Serena Montagnaro

Reviewer 2 Report

In this manuscript, the authors conducted the seroepidemiology of pseudorabies virus (PRV) in wild boars captured in the Campania Region, Southern Italy, during 2016-2017. They found that about 24% of the individuals were seropositive, indicating that PRV still circulated in the area.

I think the topic of this paper is just local but not general interest.

The authors should explain the rationale why they selected the Campania Region for the seroepidemiology.

To evaluate the risk that PRV in wild boars transmits to domestic pigs, the authors should conducted the virus detection test (e.g., PCR) as well. 

The authors should provide the information of domestic pigs in the region; for example, population, location of the pig farm, the vaccine status, the prevalence of PRV, etc.

The authors should mention how protect pigs from PRV infection in the area.

Author Response

(The authors gave the same response as above.)

Reviewer 3 Report

Authors aims to evaluate the prevalence of ADV in wild boars during the hunting season 2016 - 2017, in the Campania Region and to assess the risk factors correlated with this disease.

A very interesting work, more details should be given in terms of the multivariable logistic regression analysis, not only OR and 95% CI, but also the p-values.

  • lines 135 and 136: 302 males (60%) and 507 females (40%), but the total serum samples was 503.

At the introduction, if possible incorporate data related to risk factors that could help understand the exploration of gender and age among others.

Specific comments/suggestions can be observed highlighted in the attached .pdf file.

Author Response

(The authors gave the same response as above.)

Round 2

Reviewer 2 Report

Originally, this work was not significant. It is not novel finding that wild dears can be a reservoir of PRV. The authors had conducted the serological survey in wild dears in the Campania region 10 years ago. In this decade, it seems that no measures against PRV in wild boars has been conducted in the area. In this study, the result was similar to the previous study. Unfortunately, the authors did not adequately respond to my indications in the revised manuscript, I have to say again that the topic of this paper is just local and not general interest.

Point 1: The authors should explain the rationale why they selected the Campania Region for the seroepidemiology.

Response 1: We have selected the Campania region because it is the work area of our research group. We evaluated the seroprevalence of ADV in wild boar also in 2010, so is interesting to investigate about the prevalence of the disease in wild boar, after about 10 years. In Italy there is an eradication plan for ADV, therefore it is important to monitor wild animals to obtain an accurate snapshot of the prevalence in the region.

It is interest if the authors show a data set before and after measures against PRV in wild boars in the Campania region, but not unfortunately. In addition, the response to my point was not reflected in the revised manuscript.

Point 2: To evaluate the risk that PRV in wild boars transmits to domestic pigs, the authors should conduct the virus detection test (e.g., PCR) as well.

Response 2: The aim of this survey was to evaluate the prevalence and the changes in the prevalence of AD in wild boar. Serology is a measure of historical exposure and may or may not be linked to viremia at the time of sampling, while viral detection test (e.g. PCR) is indicated for etiological confirmation of the infection but is not suitable for estimating the prevalence of the disease.

ADV, like all herpesvirus, causes latency, therefore a negative viral detection test did not exclude the infection and can determine an underestimation of the disease.

Lastly, only serum was available for this study and this type of specimen is not appropriate for the molecular diagnosis of ADV.

I disagree to this response. PRV causes latent infection in swine, so it is important to detect the virus to evaluate the risk of transmission from wild boars to domestic pigs. The quantitative detection is even better. Indeed, there is a report that the latent PRV was quantitatively detectable using qPCR in the nervous tissue from pigs and the detection rate was high (Yoon et al, J Vet Med Sci, 68(2):143-148, 2006). In addition, not collecting tissues suitable for virus detection is considered as an inadequate preparation and is not an excuse not to do.

Point 3: The authors should provide the information of domestic pigs in the region; for example, population, location of the pig farm, the vaccine status, the prevalence of PRV, etc. The authors should mention how protect pigs from PRV infection in the area.

Response 3: Lines 72 – 86: In Italy and in all EU countries, an eradication plan is in place, including large-scale vaccination with delete-gE vaccines, application of additional guarantees for pig movements between Member States [9] with and increased efforts to control the dis-ease. Pigs can be dispatched from any Member State to any other if the conditions laid down in Decision 2008/185/EC [9] are respected. These conditions are under the con-trol of the Member State of origin and are more stringent for movements of pigs to a Member State with a higher animal health status [9]. This strategy has led to a de-crease in the incidence and prevalence of the pseudorabies in in Italy and in several European Union (EU) Member States [10].In Italy the national AD control program, which provides for mandatory vaccination with delete-gE vaccines was started in 1997 [11] and together with the Community provisions in force about AD eradication [9] resulted in a significant reduction in the incidence and prevalence of the disease, but AD has not yet been eradicated [12].In fact, currently, in Campania region, the prevalence of AD at herd and animal level is 0.5% and 0.05% respectively; unfortunately, the ADV seems to be still present in feral pigs [13].

Lines 124 - 126: Pigs are one of the most raised animals in the Campania region, in fact, there is about 22.200 pig farm, with a total of about 175.000 animals. About 96.5% of farm are small family farms, while fattening and breeding farms represent 2% and 1.5% respectively [22].

The authors should have explained the situation of domestic pigs in the Campania region, where they selected. Although they did not provide the informations about the vaccine status and the prevalence of PRV in the Campania region. Since I and other readers who do not live in Italy do not know the region, it is difficult to understand the importance of this study if the adequate information is not provided.
